# Inflammatory Biomarkers Affecting Survival Prognosis in Patients Receiving Veno-Venous ECMO for Severe COVID-19 Pneumonia

**DOI:** 10.3390/diagnostics13132203

**Published:** 2023-06-28

**Authors:** Željka Drmić, Ivan Bandić, Sonja Hleb, Andrea Kukoč, Sanja Sakan, Nataša Sojčić, Darko Kristović, Verica Mikecin, Ivana Presečki, Zrinka Šafarić Oremuš, Nikola Bradić, Jasminka Peršec, Andrej Šribar

**Affiliations:** 1Clinical Department for Anesthesiology, Reanimatology and Intensive Care Medicine, University Hospital Dubrava, Avenija Gojka Šuška 6, 10000 Zagreb, Croatia; 2Department of Health Studies, University North, 42000 Varaždin, Croatia; 3School of Dental Medicine, University of Zagreb, 10000 Zagreb, Croatia

**Keywords:** COVID-19, ECMO, survival analysis, mechanical ventilation, biomarkers

## Abstract

Severe COVID-19 pneumonia in which mechanical ventilation is unable to achieve adequate gas exchange can be treated with veno-venous ECMO, eliminating the need for aggressive mechanical ventilation which might promote ventilator-induced lung injury and increase mortality. In this retrospective observational study, 18 critically ill COVID-19 patients who were treated using V-V ECMO during an 11-month period in a tertiary COVID-19 hospital were analyzed. Biomarkers of inflammation and clinical features were compared between survivors and non-survivors. Survival rates were compared between patients receiving ECMO and propensity matched mechanically ventilated controls. There were 7 survivors and 11 non-survivors. The survivors were significantly younger, with a higher proportion of females, higher serum procalcitonin at ICU admission, and before initiation of ECMO they had significantly lower Murray scores, PaCO_2_, WBC counts, serum ferritin levels, and higher glomerular filtration rates. No significant difference in mortality was found between patients treated with ECMO compared to patients treated using conventional lung protective ventilation. Hypercapnia, leukocytosis, reduced glomerular filtration rate, and increased serum ferritin levels prior to initiation of V-V ECMO in patients with severe COVID-19 pneumonia may be early warning signs of reduced chance of survival. Further multicentric studies are needed to confirm these findings.

## 1. Introduction

The outbreak of severe acute respiratory coronavirus 2 (SARS-CoV-2) affected the healthcare system of almost every country in the world so severely that the World Health Organization (WHO) declared it to be a public health emergency of international concern on 30 January 2020, and a pandemic on 11 March 2020 [1,2]. As of January 2023, the ongoing coronavirus disease 2019 (COVID-19) pandemic had caused more than 668 million cases, with a death toll of 6.78 million globally [3].

In Croatia, as part of the national strategy against the COVID-19 pandemic, the University Hospital Dubrava was repurposed into a primary respiratory center for patients with confirmed COVID-19 infection. The primary respiratory intensive center (PRIC-IC) was a subunit in the UH Dubrava which was specialized for treating the critically ill with severe symptoms of COVID-19 that required mechanical ventilation, vasoactive hemodynamic support, continuous renal replacement therapy, and other aspects of intensive care including extracorporeal membrane oxygenation (ECMO) [4].

COVID-19 is characterized by a spectrum of diseases ranging from asymptomatic or mild cases to critical disease with acute respiratory distress syndrome (ARDS) requiring mechanical ventilation [5,6,7]. A subset of these patients continue to have refractory respiratory failure despite lung protective mechanical ventilation, prone positioning, and neuromuscular blockade, and thus require other modes of oxygenation [8,9,10,11]. Veno-venous (V-V) ECMO is an extracorporeal technique that provides oxygenation and removal of carbon dioxide from the blood via two large bore venous cannulae, a membrane oxygenator, and roller pumps, and has consequently been used as a rescue therapy in appropriate, critically ill patients [12,13]. However, it is an invasive support that brings significant risk of complications and challenges [14,15]. Management of ECMO requires the use of systemic anticoagulation to prevent patient and circuit-based thrombosis, which, in turn, increases the risk of bleeding and can have a significant impact on patient outcome [16,17]. Certain therapeutic options are available, including unfractionated heparin which is the most common drug used, but certain alternatives such as nafamostat mesylate are available [18]. During ECMO, the inflammatory response occurs as a reaction to exposure of blood to the non-endothelial surface of the extracorporeal circulation. Proinflammatory cytokine production increases rapidly which induces complement activation, directly affecting leukocytes, platelets, and the vascular endothelium. This ultimately leads to a systemic inflammatory response and eventual organ damage [19,20,21]. Acute kidney injury is common during ECMO support (21–70% incidence) and often requires renal replacement therapy which is an independent risk factor for ECMO mortality, especially with technical difficulties regarding personnel availability and anticoagulation therapy where the heparin sparing regimen that can be utilized in ECMO with heparin-coated oxygenators is often inadequate for certain CRRT filters in which there is no coating [22]. Although there is still conflicting evidence regarding whether the use of V-V ECMO is beneficial in patients with severe ARDS [23], when all the other therapeutic options are exhausted it may offer a better survival chance.

The primary goal of this study is to determine the predictive value of inflammatory response biomarkers on the survival of COVID-19 patients receiving V-V ECMO to treat severe COVID-19 pneumonia. The secondary goal of this study is to determine whether the use of V-V ECMO is associated with a decrease in intensive care unit (ICU) length of stay and in-hospital mortality compared to conventional lung protective mechanical ventilation.

## 2. Materials and Methods

This study was designed as a retrospective observational study, and it included patients admitted to the combined ICU organized in the specialized PRIC-IC UH Dubrava during a 11-month period in late 2020 and early 2021 when it was repurposed to be a national COVID hospital.

Among the 6 ICUs in the PRIC-IC, one (IC-5) was equipped and staffed with personnel experienced in treating patients receiving ECMO.

### 2.1. Patients

Patients who were ventilated < 10 days in which conventional lung protective mechanical ventilation with additional adjuvant therapeutic procedures (neuromuscular blockade and prone position ventilation) failed to achieve adequate oxygenation and/or CO_2_ removal were assessed with consilium from 2 intensivists and a cardiac surgeon to determine whether the patient was a viable V-V ECMO candidate, and if the patient fulfilled the criteria determined by the extracorporeal life support organization (ELSO) [13].

### 2.2. ECMO

Indications for V-V ECMO initiation in the studied cohort of patients were duration of mechanical ventilation < 7 days with hypoxemia defined as a PaO_2_/FiO_2_ ratio < 60 mmHg over 6 h or <50 mmHg over 3 h, or respiratory acidosis with pH < 7.2 or paCO_2_ > 80 mmHg over 6 h without a significant improvement after administration of PEEP, neuromuscular blockade, or prone position ventilation. Contraindications were irreversible neurological deficit, encephalopathy, terminal liver failure, irreversible lung changes, HIV, clinical frailty scale > 3, BMI > 40 kg/m^2^ (relative contraindication), and age > 65 (relative contraindication).

Once a decision to initiate ECMO had been made, cannulation was performed by an intensivist/surgeon team, the ECMO circuit was primed by perfusionists, and ECMO was initiated with initial settings of blood flow 60–80 mL/kg_BW_/min and fresh gas flow initially set to equal blood flow and subsequently adjusted according to the patient’s paCO_2_. After initiation of ECMO, the ventilator was set to maintain driving pressure < 15 mbar, plateau pressure < 30 mbar, tidal volume 4 mL/kg_BW_, and FiO_2_ was set at 30%. ECMO FiO_2_ was set to maintain blood oxygen saturation between 85 and 95% and was readjusted according to the patient’s serum lactate levels.

Eight patients in total received hemadsorption via a polystyrene hemadsorption kit (Cytosorb, Cytosorbents Inc., Monmouth Junction, Princeton, NJ, USA) attached to the ECMO circuit.

### 2.3. Data Collection

After institutional ethics board approval (University Hospital Dubrava ethics committee, ID 2021/2309-01), data collection was performed from the hospital information system (iBIS, IN2, Zagreb, Croatia). The recorded variables were basic demographic characteristics (gender and age), laboratory parameters at ICU admission (white blood cell count (WBC) × 10^9^/L, neutrophil and lymphocyte percentage in WBC, as well as the neutrophil/lymphocyte ratio (NLR), PaO_2_/FiO_2_ ratio (mmHg), serum D-dimer (mg/L), serum lactate (mmol/L), serum ferritin (µg/L), serum procalcitonin (PCT, ng/mL), serum C-reactive protein (CRP, mg/L), serum interleukin 6 (IL-6, pg/mL), and glomerular filtration rate (ml/min/1.73 m^2^)), basic anthropomorphic characteristics (body mass index. (BMI), kg/m^2^), presence of relevant major comorbidities (arterial hypertension, diabetes mellitus, congestive heart failure defined as NYHA status > II, chronic kidney disease defined as glomerular filtration rate < 60 mL/min/1.73 m^2^ calculated by using the CKD-EPI formula, and chronic hematologic disorders), Charlson comorbidity index (CCI), sequential organ failure assessment (SOFA) score, duration of COVID-19 disease before ICU admission, duration of ICU stay, ECMO, mechanical ventilation, and both the ICU and the in-hospital mortality rate.

The criterion for determining whether prone position ventilation was performed or not was completion of at least one session with duration over 12 h, due to the lack of evidence that there is a significant clinical benefit if sessions are shorter than 12 h [24].

### 2.4. Statistical Analysis and Propensity Matching

Continuous variables are displayed as either mean and standard deviation (SD) for values with Gaussian distribution, or median and interquartile range for data that do not follow normal distribution. Normality of distribution was assessed using the Shapiro–Wilk test. Categorical variables are displayed as counts and percentages.

Differences in continuous variables between 2 groups were tested for statistical significance using Student’s *t*-test or Wilcoxon rank-sum (Mann–Whitney U) test, depending on the distribution of data. Differences in categorical variables were tested for statistical significance using Pearson’s χ^2^ test.

Differences between groups on repeated measurements were tested for statistical significance using repeated measurement analysis of variance (RM-ANOVA) with post hoc Tukey correction.

Mortality rates and survival times between patients treated with V-V ECMO and patients who only received conventional invasive mechanical ventilation were compared. It must be noted that patients that only received mechanical ventilation were viable ECMO candidates without contraindications but were not treated with ECMO either due to the fact that, at the time of referral, there were no available ECMO machines or they were referred too late to the IC-5 coordinator (per ELSO guidelines [13]) or not at all.

In order to minimize the effect of confounding factors on the analysis of survival rates between patients who received ECMO opposed to those who only received conventional invasive mechanical ventilation, patients were propensity matched according to age, total number of comorbidities, sequential organ failure assessment (SOFA) score at ICU admission, and Charlson comorbidity index (CCI).

The Kaplan–Meier method was used to plot survival times, and survival probability between groups was tested using the Mantel–Cox log-rank test.

A *p*-value < 0.05 was considered to be statistically significant. The software packages used for statistical analysis and data visualization were jamovi v2.2.2 [25] with survminer [26] and finalfit [27] modules and JASP v0.14.1 [28].

## 3. Results

Eighteen patients were treated with ECMO during the 11-month period. There were six males and twelve females. The mean age was 53 ± 8.4 years, BMI was 30.6 ± 8.5 kg/m^2^, and the duration of ventilation before initiation of ECMO was 4.5 (1.3–6.8) days. The median SOFA score at ICU admission was 3.5 (IQR 2–4). The paO_2_/FiO_2_ ratio at ICU admission was 77 (64–105), the last recorded before initiation of ECMO (within 4 h) was 69 (49–85) mmHg, and the paCO_2_ before initiation of ECMO was 13.3 ± 5.8 kPa. The Charlson comorbidity index was 1 (IQR 1–1) and the Murray score before ECMO initiation was 3.7 (IQR 3.5–4). In terms of relevant recorded comorbidities, one patient had type 1 diabetes (5.6%), seven patients had arterial hypertension (38.9%), one patient had end-stage renal failure (5.6%), and one patient had non-malignant hematologic disease (5.6%). The duration of ECMO was 11.5 (7–19) days, while mechanical ventilation duration after weaning off ECMO was 1 (1–7) days.

At ICU admission, the recorded laboratory parameters were serum ferritin, 1143 ± 743 µg/L; D-dimer, 4.37 (0.94–4.45) mg/L; CRP, 120.7 ± 62.1 mg/L; procalcitonin, 0.16 (0.11–0.69) µg/L; IL-6, 43.5 (23.9–52.1) ng/L; WBC count, 13.5 ± 4.3 × 10^9^/L; neutrophil count, 90.9 ± 3.9%; lymphocyte count, 4.7 ± 2.3%; NLR, 20.5 (14.7–26.3); lactate, 1.99 ± 1.48 mmol/L; and GFR 110 (101–116) mL/min/1.73 m^2^.

Eleven patients died during their ICU stay and seven patients survived and were later discharged from the hospital (61.1% ICU and hospital mortality). Among the eleven patients that died, three patients died due to cannulation-related complications. All patients received remdesivir during the early course of disease and corticosteroids during the intermediate/late phases of disease, per period-specific COVID-19 treatment guidelines [29].

Survivors were significantly younger, with a higher proportion of females, significantly higher serum procalcitonin at ICU admission, and before initiation of ECMO they had significantly lower Murray scores, lower arterial partial pressure of CO_2_, lower white blood cell counts, lower serum ferritin levels, and higher GFR levels (Table 1).

After post hoc correction, significant differences in the dynamics of serum ferritin level (P_Tukey_ = 0.034, Figure 1) and WBC count (P_Tukey_ ≤ 0.001, Figure 2) were recorded between admission and the time point of ECMO initiation between survivors and non-survivors.

During ECMO treatment, there were significant differences between survivors and non-survivors in serum procalcitonin levels on Day 3 (0.3 (IQR 0.2–0.8) vs. 1.6 (IQR 1.4–4.9), *p* = 0.02), serum ferritin levels on Day 3 (567 (IQR 212–919) vs. 1156 (IQR 936–1510), *p* = 0.04), IL-6 on Day 3 (59.4 (IQR 6.9–131.9) vs. 487.6 (IQR 229.0–955.0), *p* = 0.01), and lymphocyte percentage on Day 7 (11.7 (IQR 6.8–16.5) vs. 4.2 (IQR 2.2–5.9), *p* < 0.01). No significant differences in the dynamics of change were found after post hoc correction.

Compared to the propensity matched cohort of patients treated in the same institution using conventional lung protective ventilation, there were no significant differences in baseline patient characteristics other than days elapsed since the first positive SARS-CoV-2 PCR test was obtained (Table 2). No significant differences in ICU or in-hospital mortality survival rates were found between patients receiving V-V ECMO and patients treated conventionally (Figure 3), while patients treated with ECMO had significantly higher durations of mechanical ventilation and ICU stays (Table 3).

No significant association was found between use of hemadsorption and ECMO survival (*p* = 0.38).

## 4. Discussion

The results obtained from this study of critically ill patients who received veno-venous ECMO due to severe bilateral COVID-19 pneumonia suggest that age, sex, serum procalcitonin levels at ICU admission, severity of hypercapnia, Murray score, leukocytosis, glomerular filtration rate, and serum ferritin level prior to initiation of ECMO are predictors of ICU and in-hospital survival.

The results were similar to a multicentric study by Bergman et al. [30], in which patient age and leukocytosis were also associated with increased 60-day mortality. However, unlike the results in the mentioned trial, there were no significant differences in SOFA scores and P/F ratios between survivors and non-survivors. It must be noted that the increased number of antibiotic days that was recorded in non-survivors (which directly translates to rate of bacterial superinfections) was also associated with increased mortality, which was in concordance with results published from a larger cohort from our center (from which this subset of patient was extracted) by Ćurčić et al. [31]

In a study by Shakoor et al. [32], disease severity was associated with a D-dimer cut-off of 3000 ng/mL (i.e., 3 mg/L), and patients with higher D-dimer levels were at higher risk. In our study, there were no differences in D-dimer levels between survivors and non-survivors, but patients in our study had significantly higher D-dimer levels at ICU admission (4.37, IQR 0.94–4.45 mg/L) which were probably even higher since the biochemical laboratory in our hospital does not measure values above 4.48 mg/L (if the value is greater it is displayed with an “>” operator). Stringent selection criteria for ECMO candidates and well-trained and synchronized staff have resulted in a very high survival rate, but it must also be noted that in the subgroup of patients younger than 40 years of age in our cohort, the survival rate was 100%.

Overall, ECMO mortality rate of ECMO patients in this study is in agreement (61% vs. 59%) with data published by Barbaro et al. [33] since our center falls in the “late adopting” category of COVID ECMO centers due to the fact that staff and equipment requirements were not met earlier. The reason for late adoption of ECMO in our center was the relative proximity of the University Clinic for infectious diseases which specializes in veno-venous ECMO in patients with viral pneumonia (with extensive experience acquired during the H1N1 influenza pandemic [34]); the PRIC-IC ECMO program for COVID-19 pneumonia related respiratory failure was started only after the capacity at the infectious disease clinic was exhausted.

Compared to the propensity matched controls which only received lung protective ventilation with adjuvant procedures (prone positioning and neuromuscular blockade), V-V ECMO did not achieve a reduction in mortality. It must be noted that, while an attempt was made to provide prone position ventilation to all patients that failed to achieve adequate oxygenation or CO_2_ removal using lung protective ventilation, in some patients it failed due to hemodynamic instability or worsening of arterial blood gas analyses while in prone position [10]. A meta-analysis published by Aoyama et al. supported the use of both prone positioning and V-V ECMO as adjuvant therapies added to lung protective ventilation in patients with ARDS [35].

In comparison with a study published by Cheng et al. [36], the results from both studies agree that age, severity of hypercapnia, and a lower lymphocyte count are factors that affect the survival rates of patients receiving V-V ECMO, with similar survival rates reported.

In our study, use of hemadsorption had no positive nor negative effect on survival rates in patients receiving ECMO, in contrast to results published in the CYCOV [37] trial in which it was associated with increased mortality during the first 72 h of ECMO, while the results published by Song et al. [38] suggested that there might be a survival benefit if hemadsorption was used. In general, while use of hemadsorption shows a reduction in circulating levels of interleukin 6, most trials (although performed on small sample sizes) have shown low clinical evidence regarding improvement of patient outcomes; further prospective trials are needed to establish hemadsorption as a standard therapeutic modality in COVID-19 [39].

It must be noted that conventionally treated patients who did not receive ECMO were viable ECMO candidates regarding all the criteria set by ELSO, but did not receive ECMO due to the availability of ECMO machines and the fact that some patients that were viable candidates were either referred too late (there were six ICUs working independently and three of those were staffed by intensivists from other hospitals, most of which had no previous experience with ECMO) or were not referred to ECMO ICU at all (those patients were recognized as viable ECMO candidates later, during the data collection and analysis process).

An ECMO referral protocol which would promote timely communication between ICUs should be implemented in the future.

The main limitation of this study is the sample size (18 ECMO patients); however, the number is understandable due to the scarcity of patients treated with ECMO, both due to the availability of equipment (only three ECMO machines available) and to the fact that V-V ECMO is beneficial only in carefully selected patients [13,32] and only if initiated in a timely manner [13].

Another significant limitation of this study is the fact that important ventilatory parameters such as PEEP, P_plateau_, dP, VT, and respiratory rate (all important factors in the calculation of mechanical power [40]) were not routinely recorded in the electronic medical records from which data were gathered and could not be compared between groups. Since approximately one third of the installed ventilators in the PRIC-IC did not have advanced tools to determine optimal PEEP levels (such as low-flow pressure volume loop, measurement of end-expiratory lung volume or esophageal pressure) and approximately 70% of the physicians working in the ICU during the pandemic were not experienced with lung ultrasound, ARDSNet PEEP/FiO_2_ tables [41] were used as a standard to set PEEP levels per institutional guidelines. Because of that, it is safe to say that, in most cases, PEEP levels were 15 mbar or higher (since FiO_2_ was recorded in the EMR). It should be noted that since there are different phenotypes of COVID-19 pneumonia [42] a “one size fits all” approach of setting PEEP might be suboptimal, but at the given moment it was the only option available.

## 5. Conclusions

The results of this retrospective observational study suggest that the biomarkers that might have a prognostic value in critically ill patients with COVID-19 pneumonia and severe respiratory failure treated with veno-venous ECMO are serum procalcitonin level at ICU admission, paCO_2_, serum ferritin level, white blood cell count, and glomerular filtration rate prior to initiation of ECMO. Compared to conventional lung protective mechanical ventilation, ECMO did not increase patient survival rates. Further multicentric studies that focus on the predictive value of inflammatory markers are needed to stratify which patient subgroups might benefit from V-V ECMO.

## Figures and Tables

**Figure 1 diagnostics-13-02203-f001:**
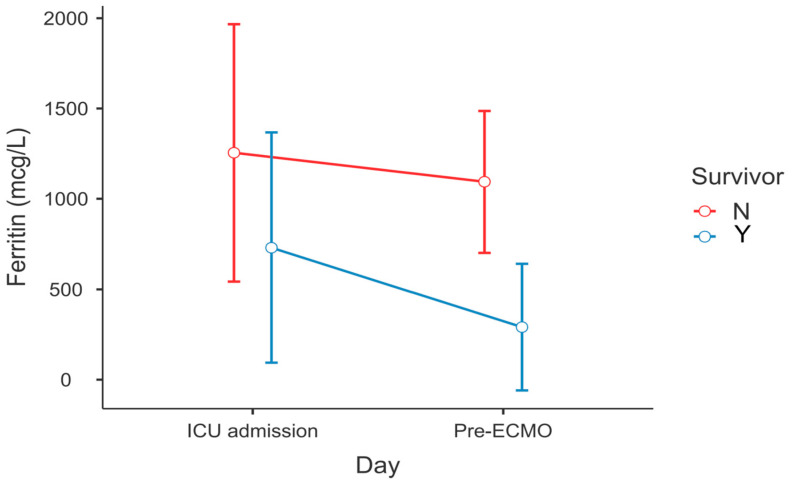
Change in serum ferritin levels between ECMO survivors and non-survivors—ICU admission vs. lab results within 6 h before ECMO cannulation (RM-ANOVA, survivors vs. non-survivors at pre-ECMO, P_Tukey_ = 0.034).

**Figure 2 diagnostics-13-02203-f002:**
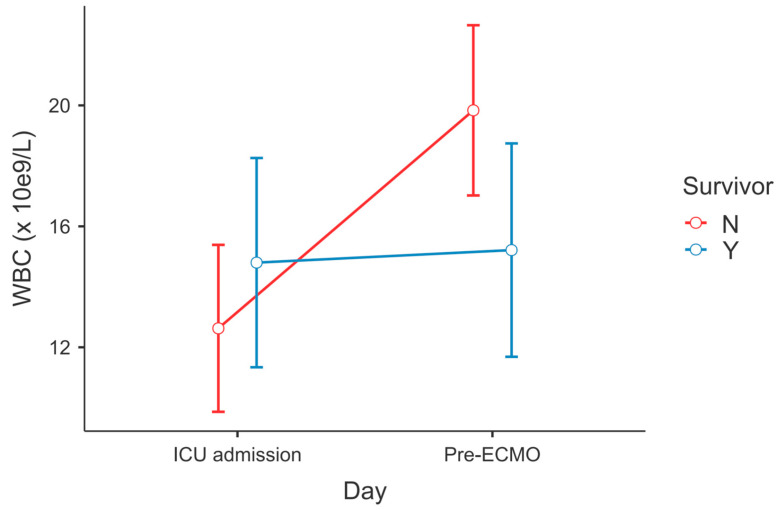
Change in WBC count between ECMO survivors and non-survivors—ICU admission vs. lab results within 6 h before ECMO cannulation (RM-ANOVA, non-survivors at admission vs. pre-ECMO, P_Tukey_ < 0.001).

**Figure 3 diagnostics-13-02203-f003:**
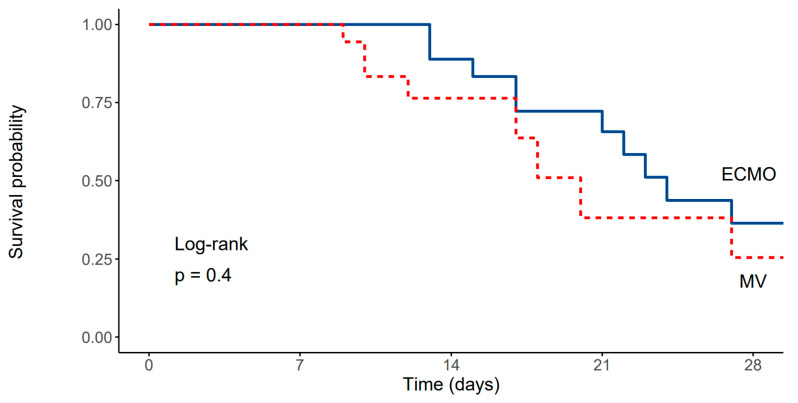
Kaplan–Meier survival curve depicting survival rates of patients treated with ECMO and the propensity matched cohort of patients ventilated using conventional lung protective mechanical ventilation (Log-rank, *p* = 0.4).

**Table 1 diagnostics-13-02203-t001:** Differences in baseline patient characteristics and markers of inflammation between ECMO survivors and non-survivors.

	Non-Survivors	Survivors	Test Statistic
	(*N* = 11)	(*N* = 7)	
**Baseline/ICU admission**			
Age	55.2 **58.0** 59.0	37.0 **47.0** 54.7	F_1,16_ = 13.18, *p* < 0.01 ^3^
BMI	25.9 **30.0** 35.3	22.7 **26.5** 33.3	F_1,8_ = 0.54, *p* = 0.48 ^3^
Sex: F/M	0.1 1/11	0.7 5/7	Χ^2^_1_ = 7.48, *p* = 0.01 ^2^
SOFA	2.2 **3.0** 4.0	2.0 **4.0** 4.0	F_1,16_ = 0.01, *p* = 0.93 ^3^
CCI	1.0 **1.0** 1.8	1.0 **1.0** 1.0	F_1,16_ = 0.34, *p* = 0.57 ^3^
P/F (mmHg)	59.0 **67.0** 109.0	71.7 **82.5** 116.1	F_1,15_ = 0.48, *p* = 0.50 ^3^
CRP (mg/L)	57.1 **101.9** 136.8	97.0 **156.9** 185.9	F_1,16_ = 3.16, *p* = 0.09 ^3^
PCT (µg/L)	0.1 **0.1** 0.3	0.2 **0.7** 2.4	F_1,15_ = 6.07, *p* = 0.03 ^3^
WBC (×10^9^/L)	10.2 **13.5** 15.0	10.8 **13.9** 20.5	F_1,16_ = 0.73, *p* = 0.41 ^3^
Neutrophil (%)	89.1 **92.8** 93.7	84.9 **91.8** 92.6	F_1,15_ = 0.81, *p* = 0.38 ^3^
Lymphocyte (%)	2.6 **4.2** 6.1	3.9 **4.6** 6.4	F_1,15_ = 0.41, *p* = 0.53 ^3^
NLR	14.8 **22.1** 36.8	14.3 **20.1** 22.0	F_1,15_ = 0.63, *p* = 0.44 ^3^
Ferritin (µg/L)	727.2 **1054.0** 1704.4	328.3 **709.5** 1794.7	F_1,12_ = 0.80, *p* = 0.39 ^3^
IL-6 (ng/L)	26.5 **50.7** 126.6	20.1 **24.5** 46.4	F_1,11_ = 1.42, *p* = 0.26 ^3^
D-dimer (mg/L)	0.8 **3.2** 4.4	1.5 **4.4** 4.5	F_1,15_ = 3.06, *p* = 0.10 ^3^
Lactate (mmol/L)	3.7 **3.7** 3.7	0.9 **1.1** 1.4	F_1,1_ = 3.00, *p* = 0.33 ^3^
GFR (mL/min/1.73 m^2^)	96.8 **108.6** 117.0	103.8 **112.3** 117.7	F_1,16_ = 0.24, *p* = 0.63 ^3^
**Treatment**			
Corticosteroids	11/11	7/7	N/A
Remdesivir	11/11	7/7	N/A
Tocilizumab	1/11	1/7	Χ^2^_1_ = 0.12, *p* = 0.73 ^2^
Hemadsorption	4/11	4/7	Χ^2^_1_ = 0.75, *p* = 0.38 ^2^
**Before ECMO initiation**			
P/F (mmHg)	43.3 **69.0** 93.3	61.2 **69.0** 81.7	F_1,16_ = 0.12, *p* = 0.73 ^3^
pCO_2_ (kPa)	13.4 **15.3** 18.7	7.4 **9.5** 10.9	F_1,16_ = 9.19, *p* = 0.01 ^3^
Murray score	3.7 **3.8** 4.0	3.3 **3.5** 3.7	F_1,16_ = 9.30, *p* = 0.01 ^3^
CRP (mg/L)	111.5 **154.8** 215.3	113.6 **127.5** 163.0	F_1,16_ = 0.39, *p* = 0.54 ^3^
PCT (µg/L)	0.4 **1.1** 4.8	0.2 **0.7** 1.3	F_1,16_ = 0.58, *p* = 0.46 ^3^
WBC (×10^9^/L)	16.5 **18.6** 22.5	13.8 **14.6** 17.8	F_1,16_ = 5.17, *p* = 0.04 ^3^
Neutrophil (%)	84.4 **89.7** 95.0	84.5 **88.9** 93.7	F_1,14_ = 0.27, *p* = 0.61 ^3^
Lymphocyte (%)	1.4 **3.3** 7.2	3.2 **4.7** 8.5	F_1,14_ = 0.62, *p* = 0.45 ^3^
NLR	12.0 **27.2** 71.0	10.6 **19.4** 28.8	F_1,14_ = 0.61, *p* = 0.45 ^3^
Ferritin (µg/L)	786.3 **1280.5** 1800.7	100.2 **352.0** 454.3	F_1,9_ = 27.00, *p* < 0.01 ^3^
IL-6 (ng/L)	83.0 **542.2** 1104.6	33.3 **53.8** 1258.0	F_1,8_ = 0.61, *p* = 0.46 ^3^
D-dimer (mg/L)	2.3 **4.3** 4.4	3.4 **4.2** 4.6	F_1,12_ = 0.33, *p* = 0.58 ^3^
Lactate (mmol/L)	1.2 **1.9** 2.8	1.5 **2.1** 2.8	F_1,8_ = 0.00, *p* = 1.00 ^3^
GFR (mL/min/1.73 m^2^)	45.4 **89.2** 100.8	104.8 **114.1** 130.1	F_1,16_ = 7.31, *p* = 0.02 ^3^

^1^ Kruskal–Wallis; ^2^ Pearson; ^3^ Wilcoxon, (Q1, **median**, Q3).

**Table 2 diagnostics-13-02203-t002:** Baseline comorbidities, laboratory values, and clinical features of patients receiving ECMO and conventional invasive mechanical ventilation.

	ECMO	Conventional IMV	*p*
	(*N* = 18)	(*N* = 20)	
Age (years)	55.5 (50–58)	53 (46.5–58)	0.557 ^3^
Gender (F/M)	6/12	8/12	0.671 ^2^
BMI (kg/m^2^)	30.6 ± 8.5	35.5 ± 6.8	0.223 ^3^
Arterial hypertension (Y/N)	7/11	10/10	0.492 ^2^
Diabetes mellitus (Y/N)	1/17	2/18	0.612 ^2^
ESRD (Y/N)	1/17	0/20	0.285 ^2^
Hematologic disease (Y/N)	1/17	0/20	0.285 ^2^
CCI	1 (1–1)	1 (1–1)	0.169 ^3^
COVID days before ICU admission	11 (9–14)	5 (3–9)	<0.001 ^3^
SOFA	3.5 (2–4)	3.5 (2–4)	0.765 ^3^
PaO_2_/FiO_2_ (mmHg)	77 (64–105)	71 (52–109)	0.604 ^3^
Ferritin (µg/L)	933 (637–1716)	622 (457–2036)	0.582 ^3^
D-dimer (mg/L)	4.37 (0.94–4.45)	4.24 (1.38–4.47)	0.324 ^3^
CRP (mg/L)	120 (91–144)	91 (66–141)	0.397 ^3^
PCT (ng/mL)	0.16 (0.11–0.69)	0.25 (0.15–0.39)	0.845 ^3^
IL-6 (pg/mL)	43.5 (23.9–52.1)	39 (21.9–58.4)	0.856 ^3^
Lactate (mmol/L)	1.99 ± 1.48	1.23 ± 0.09	0.542 ^3^
GFR (mL/min/1.73 m^2^)	109.8 (101.1–116.6)	111.9 (98.1–118.4)	0.809 ^3^
WBC (×10^9^/L)	13.5 ± 4.3	13.5 ± 5.8	0.980 ^3^
NLR	20.5 (14.7–26.3)	19.3 (13.9–26.5)	0.452 ^3^

^1^ Kruskal–Wallis; ^2^ Pearson; ^3^ Wilcoxon.

**Table 3 diagnostics-13-02203-t003:** Duration of total, pre-, and post-ECMO mechanical ventilation and ICU stay, adjuvant therapy, and ICU and in-hospital mortality of patients receiving ECMO and conventional invasive mechanical ventilation.

Variable	ECMO	Conventional IMV	*p*
	(*N* = 18)	(*N* = 20)	
Duration of pre-ECMO ventilation (days)	5.2 ± 4.5	N/A	N/A
Duration of ECMO (days)	11.5 (7–19)	N/A	N/A
Duration of post-ECMO ventilation (days)	0 (0–1)	N/A	N/A
Total duration of ventilation (days)	17 (13–26)	8 (6–14)	<0.001 ^3^
Corticosteroids	18/18	20/20	N/A
Remdesivir	18/18	20/20	N/A
Tocilizumab	2/18	2/20	0.91 ^2^
Neuromuscular blockade	18 (100%)	20 (100%)	N/A
Prone positioning	7 (38.9%)	7 (35%)	0.804 ^2^
Duration of ICU stay (days)	21.5 (17.5–29)	13 (10–17)	0.001 ^3^
ICU mortality rate	11 (61.1%)	8 (40%)	0.194 ^2^
In-hospital mortality rate	11 (61.1%)	8 (40%)	0.194 ^2^

^1^ Kruskal–Wallis; ^2^ Pearson; ^3^ Wilcoxon.

## Data Availability

Raw anonymized data in CSV format can be provided upon reasonable request.

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
