# Peer review of "Inflammatory Biomarkers Affecting Survival Prognosis in Patients Receiving Veno-Venous ECMO for Severe COVID-19 Pneumonia"

_diagnostics, 2023, doi:10.3390/diagnostics13132203_

Round 1

Reviewer 1 Report

Interesting and relevant topic is discussed, but some revisions are needed.

There are mistakes in the indexes on the first page, there are two 2, the second one should be 3.

Section Introduction. On the second page, last sentence - abbreviation ICU is better to be written with full name.

There must be one Section Materials and Methods. It should include Patients; Methods- ECMO, Statistical analysis and etc.

Page 5, paragraph 2, the text in brackets is not necessary.

Page 7, first sentence is not necessary.

Section Results, number of patients is better to be written in words. For example, page 8, line 7: “18 patients were…”.

Page 8, lines 15-16 the sentence is not clear.

Page 8, last paragraph, last sentence, description of hemadsorbstion kit is for Section Materials and Methods.

Section Discussion, page 10, last paragraph is not proper.

Author Response

Dear reviewer,   we have addressed the concerns you have expressed and revised the manuscript. We hope that you will find the revised manuscript suitable for publication.    

Reviewer 2 Report

This manuscript by Drmic et al. analyzes the role of inflammatory biomarkers on outcomes of patients receiving veno-venous ECMO for severe COVID-19 pneumonia. The concept of the study is interesting and could be valuable to readers. However, there are some minor issues to be addressed before publication.

Introduction
- When author's state that VV ECMO is: "an invasive support that brings significant risk of complications and challenges", they should specify that most of the risks are related to the use of anticoagulation (doi: 10.1016/j.thromres.2018.05.009 - doi: 10.1111/aor.14276 - doi: 10.1186/s13054-020-2726-9) and the need for continuous renal replacement therapy (doi: 10.1177/02676591211042561 - doi: 10.1111/aor.14078). Please discuss and add these five references.

Methods

- Please provide the name of your Ethical committee

- The authors stated: "Criterion for determining whether prone position ventilation was performed or not was completion of at least one session with duration over 12 hours.". Please motivate this choice and provide adequate reference.

Results

- Please include the indication and contraindication for ECMO in the methods section.

Author Response

(The authors gave the same response as above.)
